# Landslides caught on seismic networks and satellite radars

Andrea Manconi[1,2], Alessandro C. Mondini[3], and the ALPARRAY Working Group[+]

[1] Department of Earth Sciences, Engineering Geology, ETH Zurich, Switzerland
[2] now at CERC, WSL Institute for Snow and Avalanche Research SLF, Switzerland
[3] National Research Council, Istituto di Ricerca per la Protezione Idrogeologica, Perugia, Italy
[+] A full list of authors appears at the end of the paper

*Correspondence to*: Andrea Manconi (andrea.manconi@slf.ch)

**Abstract.** We present a procedure to detect landslide events by analysing in sequence data acquired from regional broadband seismic networks and spaceborne radar imagery. The combined used of these techniques is meant to exploit their complementary elements and mitigate their limitations when used singularly. To test the method, we consider a series of six slope failures associated to the Piz Cengalo rock avalanche recently occurred in the Swiss Alps, a region where we can benefit
from high spatial density and quality of seismic data, as well as from the high spatial and temporal resolution of the ESA Copernicus Sentinel-1 radar satellites. The operational implementation of the proposed approach, in combination with the future increase in availability of seismic and satellite data, can offer a new and efficient solution to build and/or expand landslide catalogues based on quantitative measurements, and thus help in hazard assessments and definition of early warning systems at regional scales.

## 1 Introduction

Landslides cause globally fatalities and devastation, with remarkable effects especially on low-income and/or developing countries (Froude and Petley, 2018). While the spatial occurrence of landslides is related to intrinsic geo-morphological, and climatic characteristics (Stead and Wolter, 2015), catastrophic failures arise when slope materials reach a critical damage state (Petley, 2004). In many cases, the ultimate trigger towards failure events is related to anthropic activities, meteorological
events, and earthquakes (Bayer et al., 2018; Huang et al., 2017; Lacroix et al., 2019).

Quantitative and accurate data on timing, location and size of landslides events are crucial to study the relationships between local and regional preconditioning factors, to recognize potential causes, as well as to identify the potential effects of climatic forcing. Moreover, efficient early warning systems at regional scale rely on the availability of accurate and complete landslide catalogues (Gariano and Guzzetti, 2016). Despite recent efforts, the knowledge on spatial and temporal landslide distribution
is incomplete. The information about landslide volume, runout, velocity, etc. is usually available only when the events threat life or damage infrastructures, as well as when they are associated with large earthquakes or exceptional meteorological occurrences. These catalogues, however, deliver only a partial picture of the impact of such events on the landscape. In addition, many landslide events are unreported because they occur in remote regions and do not have immediate and/or relevant

impacts on human activities. This strongly hinders the completeness of inventories used for hazard assessment and for calibration of early warning systems at regional scales (Guzzetti et al., 2019).

In recent years, two methods have emerged in the panorama of landslide event detection, i.e. satellite remote sensing and seismic data analyses. This is mainly due to the increased availability and quality of these datasets at global scale, as well as to the open data access policies. In particular, Earth Observation (EO) data acquired through different satellite missions are more and more exploited by systematic visual interpretation, as well as supervised and unsupervised automatic classification methodologies, in order to build catalogues of landslide events triggered by large earthquakes and/or extreme meteorological events (Mondini et al., 2019; Tanyaş et al., 2017). These methodologies strongly depend on the availability of the images, which are usually not adequate for systematic early landslide detection. Despite the identification of signatures of landslide events in seismic networks deployed for earthquake monitoring has been occasionally studied in the past (Govi et al., 2002; Weichert et al., 1994), current technical advances and diffusion of broadband seismic sensors have increased the possibility to detect and locate also landslide events of small-moderate size at regional scales. Automatic or semi-automatic procedures adapted from earthquake location routines have demonstrated fair performances (Chao et al., 2017; Dammeier et al., 2011; Ekstrom, 2006; Fuchs et al., 2018); however, while uncertainties of several km can be tolerated in case of earthquake epicentral locations, landslides are extremely confined phenomena affecting a single slope (or only small portions of it). A more accurate location of the events can be achieved with local networks specifically designed to identify mass movements (Dietze et al., 2017; Cook and Dietze, 2022). Despite, such procedures are impractical when the areas to caver are very large and the number of stations is poor, as it is typical at the scale of entire mountain chains.

In this work, we jointly use broadband seismic data and spaceborne radar imagery to show a procedure allowing for a systematic detection and location of landslides, as well as an initial definition of their area of impact, and their magnitude. We present results over the region recently affected by the Piz Cengalo, a steep granitic massive located in the central Alps at the border between Switzerland and Italy (see Figure 1), The area was repeatedly affected by large (> 1 Mm3), rock slope failure processes in the past decades, with the main event on August 23, 2017, being the largest (>3 Mm3) and most catastrophic reported in recent years, causing 8 fatalities as well as damages in the range of 50M$ (Andres and Badoux, 2018). A detailed description of the event, its preconditioning factors, potential causes, the dynamics of the rock slope failure and the subsequent debris flow reaching the village of Bondo, is beyond the scope of this work. Thus, the readers are referred to the recent literature for more information on these specific topics (Mergili et al., 2019; Walter et al., 2019).

## 2. Materials and Methods

We consider Piz Cengalo as an exemplary case to demonstrate the potential and the limits of the combination of seismic and spaceborne radar data to provide quantitative information on landslide occurrence in an alpine scenario. We benefit from the high spatial density of the AlpArray seismic network (Hetényi et al., 2018) and from the unprecedented spatial and temporal resolution of Sentinel-1 Synthetic Aperture Radar (SAR) imagery (Torres et al., 2012).

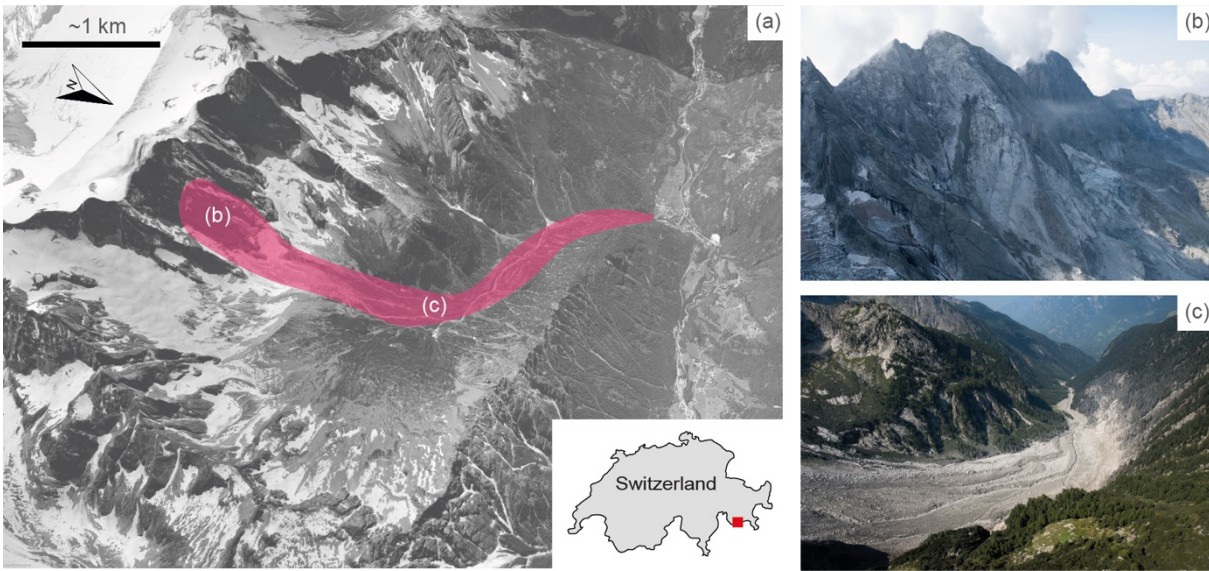

**Figure 1: Overview of the area of investigation. (a) View of the Val Bondasca, with approximate outline of the area affected by the Piz Cengalo (46.29475° N, 9.602056° E) rock avalanche and subsequent debris flows, © Google Earth 2021; (b) Detail of the release area, August 25, 2017; (c) Detail of the deposits, August 30 2017. © Photos VBS swisstopo Flugdienst.**

In the following, we describe the steps to initially define a candidate location region with seismic data, and then apply change detection investigations on Sentinel-1 SAR imagery to refine the location and identify the slope failure event. Hereafter, we will use the term "landquake" to define "landslide events recorded by seismic sensors", as increasingly proposed in literature (Chen et al., 2013). However, this term is not meant to provide additional details on specific landslide characteristics.

## 2.1 Seismic data processing

We consider a total of six events occurred at Piz Cengalo between August 21 and October 10, 2017. The landquakes are characterized by different volumes and runout, and occurred all in the same slope but at different stages of the progressive failure process: LQ1 occurred two days before the main failure, three events on August 23, 2017, (LQ2-LQ4), while LQ5 about a month later and LQ6 about two months later). Figure 2 shows the distribution of the AlpArray stations and examples of the signals for the LQ2 detected at different distances from the source. The apparent velocities are on the order of 3 km/s, thus compatible with surface waves generated by surficial mass movements (e.g., Dammeier et al., 2011). The Swiss Seismological Service (SED) routinely recognizes landslide phenomena in seismic records of stations located in Switzerland and in the vicinity of the national borders. Despite monitoring procedures are not optimized to detect mass movements, these are systematically reported. After an event detection (at least 3 stations triggered on the SED network), a first order solution is obtained by (visually) identifying coherent energy at multiple stations, typically due to S-waves, and using a regional 3D

velocity model to estimate location. In general, locations are more accurate when seismic stations are close to the event and there is good azimuthal distribution of observations. For the Piz Cengalo landquake event associated to the largest failure (LQ2), the closest station recording the event is at ~25km and the location accuracy has uncertainties on the order of ±5 km.

To perform our back analysis on the Piz Cengalo sequence, we arbitrarily define a temporal window of 10 minutes centred on the date and time provided by SED with the manual procedure described above. We consider the waveforms recorded by all the AlpArray broadband stations available for each event and focused on the HHZ channel (i.e., the vertical velocity component of high broad band sampled at or above 80Hz, generally 100 or 200 Hz). The choice of the HHZ channel is justified by previous studies showing that such component usually entails the largest energy in case of landquakes (e.g., Dammeier et al., 2011).

We apply a STA/LTA detection (see details and parameters in the Supporting Information, table S1) to find the onset time of the event at each station. Then, we compute the time delay between the first triggered station (i.e., the first station recording an event, assumed to be the closest to the event) and all the other stations identifying an event in the same temporal window. The resulting values are interpolated on a regular grid of 0.25 x 0.25 degrees, spatially smoothed with an average filter (3x3 kernel), and then normalized to obtain a new function defined here as "Likelihood of Landquake Location" (LLL). The

candidate region of interest (ROI) potentially affected by a landquake is defined by considering LLL>0.95, and to constrain the change detection processing on a spatial subset of available Sentinel-1 radar scenes.

**2.2 Sentinel-1 SAR data processing**

We adopt the change detection approach proposed in (Mondini, 2017), here specifically modified to tackle single events instead of populations of landslides. The analysis is performed to identify potential variations of surface backscattering occurred

between the pre- and post-event images, over the area with LLL>0.95. In fact, changes of the radar brightness coefficient (Beta Nought, $\beta_0$) have demonstrated to be a suitable indicator for the detection of landslide events of different size and occurred in different geographic scenarios (Mondini et al., 2019). In the maps recording the temporal changes of $\beta_0$, landslides appear as clusters of similar values in a bulk of speckles. After data acquisition, pre- and post-event Sentinel-1 radar imagery require the following steps: radiometric and geometric corrections, multi-looking, filtering of the intensity values, co-registration (pair

alignment), computation of changes (logarithm of the $\beta_0$ ratio), and ellipsoid correction. The result is a $\beta_0$ changes map with a cell resolution of about 14 x 14 m. Further, the $\beta_0$ changes map is segmented using a parametric watershed approach (Roerdink and Meijster, 2000) in which the scale level and the moving window kernel size parameters of the intensity algorithm are automatically assigned minimizing a cost function (Mondini, 2017). The segmentation process is aimed at identifying a few large segments (e.g., the largest, potentially delineating changes associated to the landquake) in the candidate area LLL>0.95,

and a number of small segments intercepting the speckle-like effect. Thus, the landquake event can be recognized as an outlier in the segment's distribution of the areas. The boundaries of the outlier segment provide the potential location of the landquake.

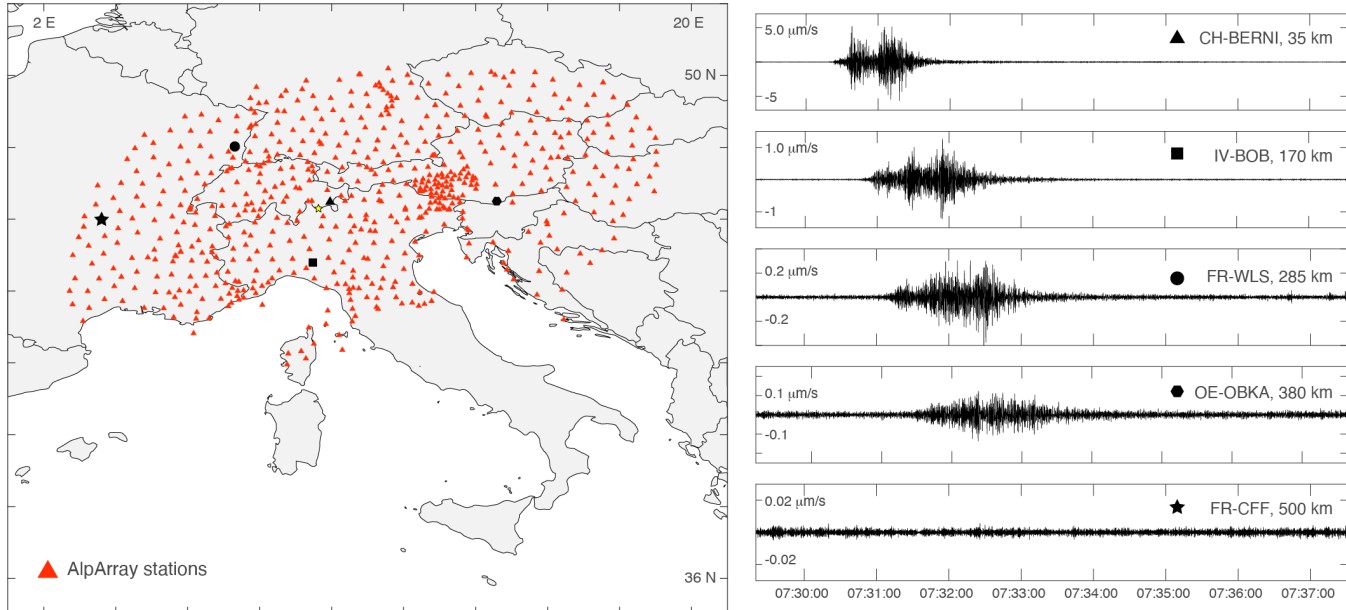

Figure 2. Seismic network and data. (left) The AlpArray network of broad band stations (red triangles). (right) Selected signals (vertical component HHZ) recorded by AlpArray stations located at different distances from event LQ2 (see table 1), occurred on August 23, 2017 (i.e., the main Piz Cengalo rock avalanche event, yellow star in the map).

| Event ID | Date/Time (UTC) | ML | MD* | ML/MD | Vol (Mm³) |
|---|---|---|---|---|---|
| LQ1 | 2017-08-21T09:29:09 | 2.3 | 3.03 | 0.75 | 0.078 - 0.167 |
| LQ2 | 2017-08-23T07:30:27 | 3.0 | 3.71 | 0.80 | 1.65 - 2.61 |
| LQ3 | 2017-08-23T09:03:57 | 1.3 | 2.86 | 0.45 | 0.02 - 0.14 |
| LQ4 | 2017-08-23T09:36:16 | 2.1 | 3.22 | 0.65 | 0.12 - 0.50 |
| LQ5 | 2017-09-15T20:04:36 | 2.3 | 3.26 | 0.70 | 0.23 - 0.41 |
| LQ6 | 2017-10-10T02:58:41 | 1.1 | 2.65 | 0.41 | 0.014 - 0.035 |

**Table 1. Summary of the landquakes analysed in this study and associated to the Piz Cengalo slope failure. ML are estimated by SED, while average magnitude duration (MD) and volumes are computed following Manconi et al., 2016, by considering the event duration on all triggered AlpArray stations. Note that all LQ events have ML/MD less or equal to 0.8, i.e. they can be discerned from earthquake events which typically have ML/MD ~ 1.**

## 3. Results

To distinguish between local earthquakes and landquakes, we applied the method proposed in (Manconi et al., 2016), based on the ratio between the local magnitude and the duration magnitude (see Table 1). This method classifies the Piz Cengalo sequence as landquakes. Moreover, the volumes computed by considering the empirical relationship with the duration magnitude (Manconi et al., 2016) are in agreement with the ones measured with LiDAR (see Walter et al., 2019).

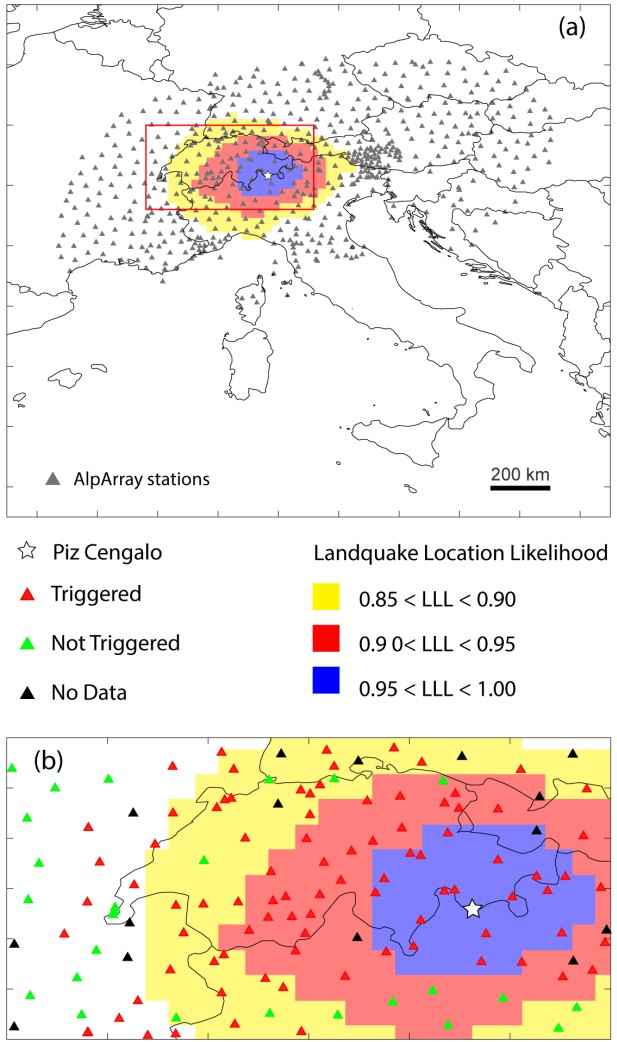

Figure 3. Likelihood of Landquake Location (LLL) based on the arrival time of seismic signals recorded by AlpArray stations. This basic analysis of the seismic data is used to constrain the approximate location where a landslide event has occurred. (a) LLL over the entire AlpArray network (b) Zoom on the areas with high likelihood. The area 0.95<LLL<1.0 is used to confine the change detection analysis. True location of the Piz Cengalo event (white star) is also shown.

Figure 3 shows the results obtained by analysing the seismic data available for the LQ2 event. This is the largest landquake, and its seismic signature was detected by tens of stations up to ~400 km distance from the source (see also Supporting Information). The computed LLL function is approximately centred on Piz Cengalo massive. The area within LLL>0.95 is in the order of 35,000 km$^2$, i.e., ~2% of the entire grid considered in the interpolation. However, this area is still very large for an accurate identification of a slope failure event affecting an area of about 1 km$^2$ (Walter et al., 2019). The initial candidate

region defined by the LLL function is used to first identify the available Sentinel-1 imagery in terms of time of acquisition and orbit. In this specific case, the suitable Sentinel-1 orbits are the T015, ascending, and T066, descending, respectively. Then, the change detection processing is not applied to the entire image, but only to the area with LLL>0.95, which is 20% of the acquired SAR scene.

Figure 4 shows the results of the change detection analysis obtained on the ascending T015 imagery (see Supporting Information, Table S2). Due to the temporal proximity of the LQ1-LQ4 sequence (occurred within two days, see Table 1), the events cannot be singularly discriminated, because the Sentinel-1 constellation (when both Sentinel-1A and 1B are operative) revisit time in Europe was of six days in the period of analysis. The LQ2, however, has been certainly the main cause of the surface changes, and for this reason we refer hereafter mainly to this event. The outlier segment covers an area of ~0.9 km$^2$,

about two orders of magnitude larger than the average areas of the segment's distribution. Moreover, the segment is elongated and has thus a very low value of roundness (defined the area of the circle with the same length as the polygon to the polygon area, see results in the scatter plot in Figure 4, bottom panels). The footprint and the dimensions of the segment identified are in very good agreement with the area affected by the rock avalanche (Walter et al., 2019). Since the events LQ5 and LQ6 are smaller in magnitude compared to the LQ1-LQ4 sequence, the changes on the SAR image cannot be univocally and

automatically defined as for the LQ2 (see Figure 4 and Supporting Information, Figure S2 and S3). In fact, the LQ5 event is likely the fifth in size segment in the region analysed, with a quite stretched/elongated shape, and a roundness value of 0.17 (median roundness equal to 0.47, and first quartile equal to 0.37). The first and the third segments are contiguous, rounded the first, and amorphous the third, indicating a large change of ~21.4 km$^2$, about 25 kilometres north Piz Cengalo, in an area where landslides were not reported. The second and the fourth segments, contiguous and rounded as well, record a change of ~15.3

km$^2$ over the 'Cima di Castello' mountain, 6 Km east Piz Cengalo valley. Their roundness ranges from 1.6 (for the amorphous segment) to 0.45. The LQ6 event does not produce statistically relevant changes in terms of size and in terms of shape.

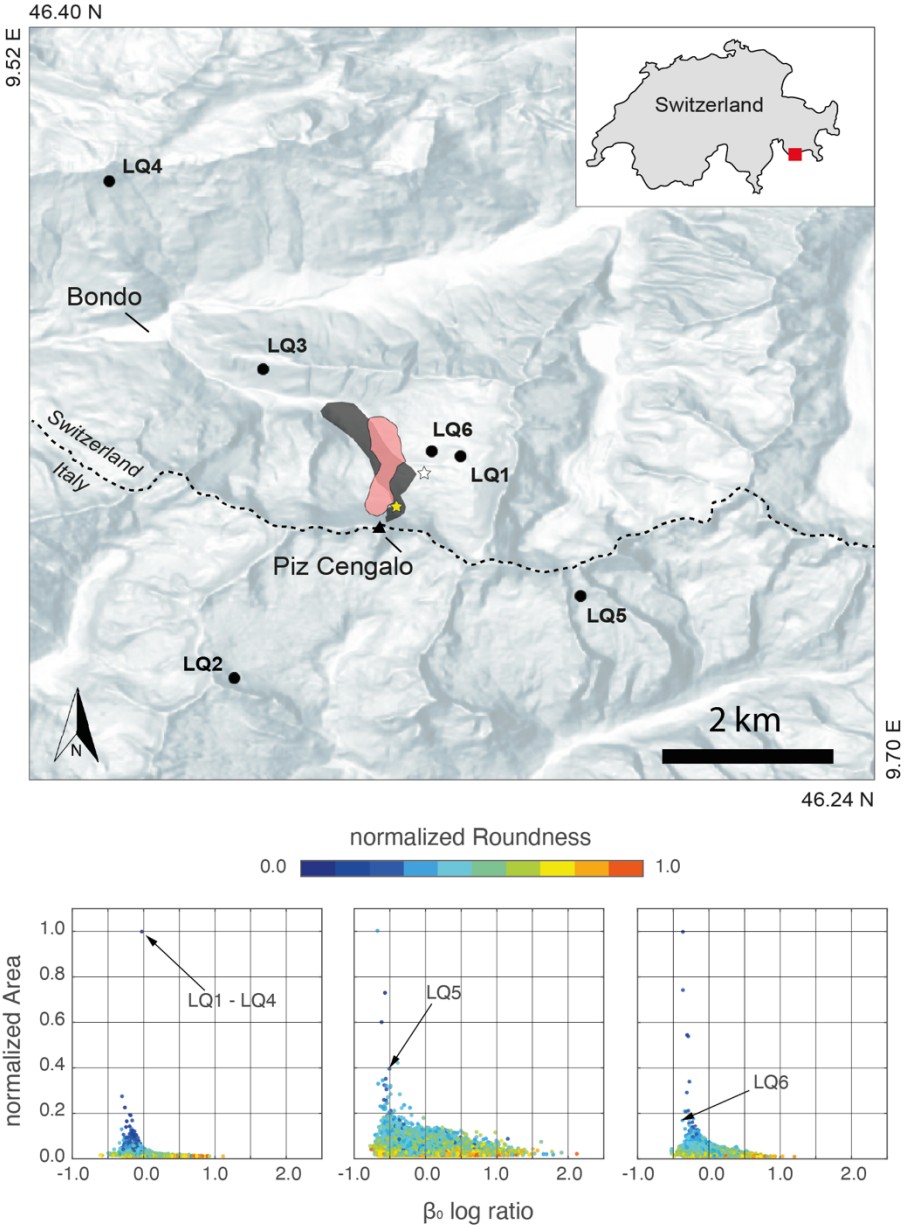

**Figure 4. (top)** Results of the change detection analysis. The red polygon shows the area identified as potential landquake location for the main Landquake event (i.e., LQ1-LQ4) identified by processing the Sentinel-1 pre- and post-event, while the grey polygon is the area hit by the rock avalanche (cf. Walter et al., 2019). The white star and the yellow star show the locations of the largest segments for LQ5 and LQ6, respectively, identified within the Bondasca valley. See also Supporting Information for more details on the segmentation results. The black dots show the epicentral locations provided by SED (see Table 1). **(bottom)** Scatter plots showing the distribution of changes in radar backscatter versus normalized areas of the segments. The colour map shows the normalized roundness. The segments associated to the Landquakes are identified by the black arrows. See the text for more details.

## 4. Discussion

Seismic data are capable to provide an indirect evidence of the time of landslide occurrence also in inaccessible locations, but independent verification of the location is necessary for event confirmation and classification (Ekström and Stark, 2013). On the other hand, remote sensing data can deliver direct evidence of the areas hit by landslide events, but independent observations are necessary to identify the exact time of occurrence (Guzzetti et al., 2012).

We propose here an approach exploiting seismic and remote sensing (specifically, space borne SAR data), which is suitable for the development of automatic pipelines aimed at a systematic identification, location and first evaluation of landslide events. We have shown as an exemplary case the application to a sequence of events recently occurred in the Swiss Alps. Our results provide several hints on the potential application of this approach in operational scenarios. We have applied a STA/LTA approach for the identification of the event on an arbitrary constrained temporal window. The STA/LTA method has shown to be suitable for the automatic detection of mass movements in continuous seismic records also for early warning purposes, although specific calibration of the parameters used is necessary and depend on the sensors, the network configuration, and local conditions (Coviello et al., 2019). One of the main arguments against the use of the STA/LTA approach in the detection of mass movement signals lies in the inaccuracy for the determination of the event's onset, which might cause errors on the subsequent location procedures (Fuchs et al., 2018). Since we refine the location using the remote sensing imagery, the STA/LTA approach is sufficient to constrain the candidate region for the change detection task. Inaccuracies up to few seconds of the STA/LTA detection that would cause large inaccuracies in location routines based on seismic data only, would cause only negligible changes on the LLL function.

An important problem after the detection of an event is the distinction and/or classification of the signals recorded in continuous seismic waveforms (e.g., earthquakes, explosion, mass movements, anthropic sources, etc.). Several authors proposed empirical based relationships, signal processing and/or or machine learning strategies, achieving good performances (Dammeier et al., 2016; Hibert et al., 2014; Moore et al., 2017). The approach used here, proposed by Manconi et al., 2016, shows that in the Piz Cengalo sequence all events could have been classified as landquakes. Moreover, the evaluation of the rockslide volumes based on the empirical relationship observed with the duration magnitude provided good agreement with independent volume measurements. The same approach has been recently implemented in an operational regional system in Taiwan showing encouraging results (Chang et al., 2020).

Despite the candidate location is identified with a basic proximity approach, the source region is already reasonably well constrained for all six LQ events considered (see also Supporting Information, Figure S1). This result is possible only when a relatively high spatial density of seismic sensors is available, such as the AlpArray network. We tested how the estimation of the area LLL>0.95 would vary by omitting the nearest seismic stations (see table 2). For LQ1 and LQ2, there is a clear increase of the LLL>0.95 area, although the values in percentage with respect to the entire size of the network still show a substantial benefit in the reduction of the candidate area for the subsequent change detection analysis. In case of smaller events, the differences are of difficult evaluation because the number of stations operative at the moment of the event, the signal/noise

ratio of the seismic data, as well as the grid interpolation step have a more relevant effect. The results of this test show that the size of the area for initial guess of landquake location is thus not only related to the distribution of the stations, but also to the size (and likely the dynamic) of the event. At this stage we cannot provide general network requirements that could be exported in other case scenarios. More advanced location routines can be applied, but homogenization of procedures across large areas like entire alpine chain is not straightforward. In addition, an increased level of complexity would not certainly correspond to

an increase of accuracies for landslide location.

| Event ID | All network<br>km$^2$ x 10$^4$ (%) | Case 1<br>km$^2$ x 10$^4$ (%) | Case 2<br>km$^2$ x 10$^4$ (%) |
|---|---|---|---|
| LQ1 | 1.24 (0.7) | 2.86 (1.7) | - |
| LQ2 | 3.55 (2.1) | 6.03 (3.6) | 10.3 |
| LQ3 | 0.85 (0.5) | 0.7 (0.4) | - |
| LQ4 | 2.70 (1.6) | 2.63 (1.56) | - |
| LQ5 | 2.24 (1.3) | 2.63 (1.56) | - |
| LQ6 | 1.08 (0.64) | - | - |

**Table 2. Results of the assessment of differences between areas with LLL>0.95 when reducing the number of stations used for the**
**interpolation. In case 1 we removed all stations triggered between 0 and 5 seconds, i.e., up to ~50 km from the event. In case 2 we removed stations triggered between 0 and 10 seconds, i.e., up to ~65 km. Missing values are because of lack of stations for reliable interpolation. Values in percentage are computed with respect to the entire interpolation grid, i.e., 1.71 x 10$^6$ km$^2$.**

As far as the change detection analysis on the Sentinel-1 SAR data is concerned, the location of landquakes as the LQ2 (i.e. in this case the LQ1-LQ4 sequence) is straightforward. The event was large and caused a vast drop of the backscattering coefficient in the post event image, spatially over sizing the surrounding random changes always present in SAR images (speckling-like effect). Furthermore, other environmental changes in the area are not relevant, and in this specific case, mostly in the direction of an increase of the backscattering coefficient. The results of the segmentation are unambiguous in all the

images whatever the acquisition mode and the polarization are, even if the final segments can be slightly different. Additionally, post processing, like smoothing or gap-filling filtering, can also change partially the final shape of the segment and the identified area. On the contrary, the identification of the LQ5 and LQ6 events shows more complexity and it is not straightforward. Their signals emerge only in the ascending imagery with VH polarization, a possible indication of a weak change of roughness along the slope (Sung and Holzer, 1976). According to seismic data, their sizes are much smaller

compared to LQ2, and then the corresponding changes of the backscattering coefficient are expected to be less distinguishable

in the bulk of random speckles (see Figure 4). In fact, when the signs left in the $\beta_0$ changes map have a size comparable to other environmental changes, or the speckling-like segments, landslides cannot be univocally recognized. Regarding LQ5, only a supervised post processing, including customised filtering to facilitate the segmentation, and manual cluster shape analysis (Mondini et al., 2019) over the valley allowed highlighting a potential segment of interest. The segment is the fifth in

size among millions, with a stretched shape compatible with the slope process under study. On the contrary, the first four clusters are rounded, or amorphous, and more adequate to represent other types of processes occurred in the area. Multi-band, multi-polarisation data, and a shorter revisit time of the satellite would have probably helped in reducing the environmental noise and then in surfacing the segment. For LQ6, a small but clear signal along the slope is present in the catchment, but is not large considering the entire distribution of segments. Other geometrical parameters, such as for example the elongation or

the roundness of the segment's area, do not help (see Figure 4). We can then assert that LQ6 is below the limit of the spatial resolution of the used images. A potential adaption for the operational implementation of our approach could be running the change detection task on progressively increasing LLL thresholds (e.g., 0.95, 0.975, etc.). This could provide additional hints on possible hot-spots, which can be verified with subsequent SAR acquisitions and/or supplementary remote sensing imagery (space-borne or air-borne).

In table 3, we summarize the strong points and the shortcomings of the herein proposed methodology, including the working hypotheses to overcome the current limitations. One of the main advantages of the pipeline is that the data pre-processing can be fully automatized. Moreover, some of the limitations depend mainly on technical constraints that might be overcome in the near future. There are a number of problems, however, that depend on the intrinsic limitations of the data considered and it is difficult at the current stage to have a clear definition of the best suited strategies to improve the performance. We will perform

future evaluations on the continuous processing of seismic data to make a substantial assessment of the potential implementation of our procedure, to find the best compromise for specific parameters, as well as to minimize false/missed alarms.

| Task | Proposed approach | Pros | Cons | Comments |
|---|---|---|---|---|
| Event detection | STA/LTA | Straightforward implementation | Pre-Calibration of the parameters is needed, inaccuracies in the determination of the event's onset | Minor impacts in a procedure that relies on additional data for a refinement of the location |
| Landquake classification | ML / MD < 0.8 | Performance already tested in alpine contexts | Large variabilities due to different approaches for the estimation of local and duration magnitudes | Ad-hoc relationships can be calibrated over smaller areas for better accuracies |

| | | | | |
|---|---|---|---|---|
| Preliminary location | LLL > 0.95 | Straightforward implementation | Depends on stations distribution, grid step, and interpolation method | Minor impacts in a procedure that relies on additional data for a refinement of the location |
| Location refinement (1) | Satellite Radar imagery | Day/night, any weather conditions | Relatively low spatial and temporal resolution, geometric distortions | High resolution data will be available in near future, according to the space missions planned. To complement with optical imagery when available |
| Location refinement (2) | Optical Imagery | Identification of polygons vs. pixelwise analysis | Ambiguity when events and/or associated changes in SAR imagery are too small | Benefit from the use of higher resolution imagery and/or acquired in different bands |


**Table 3. Summary of the pros and cons related to the herein proposed strategy for landquake identification, location, and classification.**

## 5. Conclusions

The key take-home message of this study is to show how the systematic combination of seismic and remote sensing data can
be useful for identification and mapping of landslide events. The use of SAR satellites shows the advantages of all weather, day and night, and systematic acquisitions at global scale. When available, optical imagery and/or SAR imagery acquired with different bands, full polarimetric, or with higher spatial resolution can eventually contribute to increase the quality and the quantity of the information. We believe that combining seismic and spaceborne data is a viable approach for a future operational monitoring system at the scale of the Alps, and for this reason this work can be the starting point to raise awareness
in the community, as well as to foster cooperation and the funding necessary for such an endeavour. We conclude remarking that our approach is intended to be used for systematically populate landslide catalogues relying on quantitative and accurate information on timing, magnitude and frequency, also in remote areas. Improved catalogue completeness is very important for the calibration of regional early warning systems based on rainfall thresholds, as well as on regional hazard assessments (Guzzetti et al., 2019). An increase of the availability of remote sensing imagery with daily or sub-daily revisit times could
lead to an employment in early detection of landslide events and possibly in disaster response scenarios, but these potential applications have to be evaluated in future studies.

## Team List

Components of the AlpArray Working group (23 January 2021, see details at: http://www.alparray.ethz.ch/):

György HETÉNYI, Rafael ABREU, Ivo ALLEGRETTI, Maria-Theresia APOLONER, Coralie AUBERT, Simon BESANÇON, Maxime BÈS DE BERC, Götz BOKELMANN, Didier BRUNEL, Marco CAPELLO, Martina ČARMAN, Adriano CAVALIERE, Jérôme CHÈZE, Claudio CHIARABBA, John CLINTON, Glenn COUGOULAT, Wayne C. CRAWFORD, Luigia CRISTIANO, Tibor CZIFRA, Ezio D'ALEMA, Stefania DANESI, Romuald DANIEL, Anke DANNOWSKI, Iva DASOVIĆ, Anne DESCHAMPS, Jean-Xavier DESSA, Cécile DOUBRE, Sven EGDORF, ETHZ-SED Electronics Lab, Tomislav FIKET, Kasper FISCHER, Wolfgang FRIEDERICH, Florian FUCHS, Sigward FUNKE, Domenico GIARDINI, Aladino GOVONI, Zoltán GRÁCZER, Gidera GRÖSCHL, Stefan HEIMERS, Ben HEIT, Davorka HERAK, Marijan HERAK, Johann HUBER, Dejan JARIĆ, Petr JEDLIČKA, Yan JIA, Hélène JUND, Edi KISSLING, Stefan KLINGEN, Bernhard KLOTZ, Petr KOLÍNSKÝ, Heidrun KOPP, Michael KORN, Josef KOTEK, Lothar KÜHNE, Krešo KUK, Dietrich LANGE, Jürgen LOOS, Sara LOVATI, Deny MALENGROS, Lucia MARGHERITI, Christophe MARON, Xavier MARTIN, Marco MASSA, Francesco MAZZARINI, Thomas MEIER, Laurent MÉTRAL, Irene MOLINARI, Milena MORETTI, Anna NARDI, Jurij PAHOR, Anne PAUL, Catherine PÉQUEGNAT, Daniel PETERSEN, Damiano PESARESI, Davide PICCININI, Claudia PIROMALLO, Thomas PLENEFISCH, Jaroslava PLOMEROVÁ, Silvia PONDRELLI, Snježan PREVOLNIK, Roman RACINE, Marc RÉGNIER, Miriam REISS, Joachim RITTER, Georg RÜMPKER, Simone SALIMBENI, Marco SANTULIN, Werner SCHERER, Sven SCHIPPKUS, Detlef SCHULTE-KORTNACK, Vesna ŠIPKA, Stefano SOLARINO, Daniele SPALLAROSSA, Kathrin SPIEKER, Josip STIPČEVIĆ, Angelo STROLLO, Bálint SÜLE, Gyöngyvér SZANYI, Eszter SZŰCS, Christine THOMAS, Martin THORWART, Frederik TILMANN, Stefan UEDING, Massimiliano VALLOCCHIA, Luděk VECSEY, René VOIGT, Joachim WASSERMANN, Zoltán WÉBER, Christian WEIDLE, Viktor WESZTERGOM, Gauthier WEYLAND, Stefan WIEMER, Felix WOLF, David WOLYNIEC, Thomas ZIEKE, Mladen ŽIVČIĆ, Helena ŽLEBČÍKOVÁ.

## Author Contribution

AM and ACM: Conceptualization, data analysis, manuscript writing and revision. ALPARRAY working group: Seismic data collection and maintenance.

## Competing interests

The authors declare that they have no conflict of interest.

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
