# Peer review of "Landslides caught on seismic networks and satellite radars"

_Natural Hazards and Earth System Sciences, 2022_

## Author Response (AR1)

The paper "Landslides caught on seismic networks and satellite radars" discusses the possibility to apply an integrated approach which combines broadband seismic data and satellite images for detecting landslides over large areas. To this aim, the approach was tested with a rock avalanche event occurred in Central Alps on 23rd August 2017.

GENERAL COMMENTS

I really appreciated the content of this paper, which is also fairly well-written. Although the proposed approach can be roughly summarized as a combination of two already-existing methodologies (i.e., Manconi et al., 2016 for seismic data processing and Mondini, 2017 for satellite imagery analysis), the manuscript presents several innovative elements which certainly fit with the aim of the Journal. However, before the article can be accepted for publication on NHESS, several aspects of the work must be improved to clarify the obtained results and substantiate the novelty of the proposed approach.

**We thank the reviewer #1 for this positive feedback. Here below we reply point by point to the comments and concerns.**

SPECIFIC COMMENTS

1) Structure of the work: the authors should better distinguish the "Materials and methods" section and the "results" section. In the current form of the manuscript, there is a bit overlap between the sections (e.g., Table 1 should be included in Results)

**We will modify accordingly to avoid overlap between Methods and Results. Table 1 has been moved to the Results section.**

2) Seismic data processing: the authors should better clarify "the step forward" of their approach with respect to the method employed by Swiss Seismological Service (SED) for detecting landslide phenomena. It seems that the identification of the candidate area is strongly related to an arbitrary constrained temporal window which, in turn, depends on the outcome of SED approach

**The step forward of our approach is the possibility to automatize the processing pipeline, because the current SED procedure for the location of the mass movements is manual/visual and thus arbitrary. The temporal window selected for this demonstrative case is arbitrary. Our idea is to perform future evaluation on the continuous processing of seismic data in order to find the best parameters, to minimize false alarms and make a substantial evaluation of the potential implementation of such procedure. But this is beyond the scope of the current manuscript.**

3) Analysis of the obtained results: the authors should perform a more in-depth analysis of the obtained results. At present, the "results" section is a bit lacking and several aspects are not investigated at all. Just two examples:

• six landquakes have been introduced in the first part of the work but, in practice, only LQ2 is considered (LQ5 and LQ6 are just partially addressed). Are the predicted volumes consistent with real ones?

**The volumes predicted with the empirical evaluation of the seismic waves are in agreement with the ones measured with LiDAR, we compared with the values published by Walter et al., 2019. The difficulty to evaluate the real accuracy of estimated volumes is that quantitative measurements are rare, and in most cases not performed after each single event.**

• Are there other areas which show surface changes after satellite imagery analysis? If so, it would be important to investigate this point for better understanding the reliability of the proposed approach.

**Yes there are. In fact, the surface backscatter, so as measured in the microwave range by SAR images with fixed acquisition geometries, can experience several types of changes including: 1) changes due to a land cover change (e.g. from vegetation to bare soil, eventually caused by the occurrence of landslides), 2) changes in the soil moisture content (see dielectric constant) or soil roughness, and 3) a sort of random changes caused by the speckling-like effect that can affect wide portions of the single images in particular in vegetated areas. These changes can have intensities, sizes (area) and shapes quite different. In particular, landslides appear in the measures of backscatter changes as clusters of similar pixels (all dark or all light), with elongated shapes in a bulk of "salt-and-pepper noise". These elements potentially allow for an automatic or manual (interpretation) landslide recognition among other changes. In our case, LQ2 is very large compared to other changes for at least two reasons; 1) it's a big landslide, 2) there are no other significant changes around the cluster left by the landslide (quite a static situation except for the landslide occurrence). LQ5 is a much smaller event compared to LQ2 and left a sign which is quite clear (dark), elongated, but just a bit larger than the typical clustering of the salt and pepper noise, and smaller than other changes left by a dynamic evolution of the surrounding environment (see snow processes). In this case, the analysis of the area shows that the landslide segment is the largest in the "Bondo valley" but at least four other segments are relevant (in terms of area). The landslide can be recognised (for the moment manually) through the interpretation of the elongated shape and through its position (e.g. it is not in a flat area). As far as LQ6 is concerned, this episode probably determines a lower limit for the technique: in the "Bondo valley" there is a clear dark cluster, located in a position consistent with a landslide, but its shape is not elongated (the landslide is too small for the pixel size of the used image) and its size is comparable to the size of the bulk. LQ5 & LQ6 can be seen as red flags alerted by the procedure that need further investigation (optical, field survey, or higher SAR images resolution).**

**Results and discussion have been changed accordingly:**

In my opinion, the authors should rely on the material included in the supplementary files for improving the analysis of the obtained results

**Thanks for this suggestion. We have added additional plots on the Figure 4 to cope with this request.**

4)   I suggest to slightly modify the "Discussions" section in order to clarify strong points and shortcomings of the proposed approach. In my opinion, a table which summarizes these aspects would help in this regard

**Thanks for this suggestion, we have developed the table 3.**

TECHNICAL CORRECTIONS

Caption Table 1: "have ML/MD" is repeated twice

**Thanks, corrected.**

Figure 2: please add on the map the location of Piz Cengalo

**Thanks, added.**

Citation: https://doi.org/10.5194/nhess-2022-34-RC1

The paper shows, thorouhg a case study, the potentiality of integrating seismic and EO data to improve landslide mapping capabilities. The proposed approach uses broadban seismic networks to detect landslide events and SAR imagery to spatially locate the event.

The paper is well written and well organized. The results shown are promising. I think that the parper is worth to be published. However I would propose to improve the discussion section. I agree with referee 1 on clarifying stron and shortcommings.

**We thank reviewer 2 for the positive feedback. Here below we reply point by point to the comments and concerns.**

COMMENTS

1)   Proposed approach: It is understood that the method strongly depends on the quality/density of the seismic network. That means that nowadays it is hardly scalable to other places where landslides are a major issue. I wonder if there could be the possibility to analyze the network requirements". I mean have you tested not to use all the seismometers and just see how much the preliminary location decrease  as a function of the number and density of the used seismometers? This could be a good output of the paper.

**We thank the reviewer; this is a very valuable comment. It is difficult to provide minimum requirements for the seismic network, because the capacity of detection depends on many aspects, including size and type of the event, number of stations operative at the moment of the event, signal/noise ratio of the seismic data at the moment of the event. Considering the suggestion of the reviewer, we have calculated how the area of the function LLL>0.95 (preliminary location) would change if the nearest triggered stations would not be considered in the calculation. We have included a table and additional discussion for this specific point.**

2)      As stated by the authors, LQ5 and LQ6 detection is ambiguous and strongly depends on the user. I understand that the authors are refering  here to Sentinel-1 data. How important is here the resolution or the number of images important? It would be nice to mention it in the work.

**This question is highly relevant: the adequacy of the resolution of the images used to detect landslides is probably one of the main discussed topics in geomorphology, also when optical imagery is used. In our case, the resolution of Sentinel 1 hampers for sure the possibility of a certain detection of LQ5, and LQ6 (LQ6 in particular) because the two events are very small, and the changes left, in terms of size are, quite similar to the changes left by the salt-and pepper noise (LQ6), or other changes occurring not too far from the 'Bondo valley' (LQ5). In these cases, the landslide detection cannot be entrusted to the sole use of the area but, when possible, it must make use of other geomorphological constraints, including shape and geoenvironmental factors (e.g., slope). It should also be remarked that, when SAR is used, there are other factors that should be taken into account, in particular the relative**

geometries between satellite and slope where the landslide occurs: big landslides in very unfavorables geometries can be seen much smaller, and/or remain undetected.

**The quality of the detection in LQ5 would probably benefit from a multitemporal analysis because the environment in between the two images used to measure the change of the backscattering was quite dynamic (snow?), giving origin to several clusters of changes. Arguably, in our work we decided to use a bi-temporal approach in which, systematically, the pre-event image was the last image acquired before the landquake, and the post-event image was the first acquired after the landquake to privilege the rapid detection. In this case, results could have been improved with a shorter revisit time of the satellite to limit the occurrence of changes not related to the occurrence of the landquakes. Other, and more effective mapping techniques might be linked in cascade where our procedure flags areas of changes like in LQ5 & LQ6.**

**Further information can be found in table 3 (pro. & cons) not reported here, and in the modified results section. The discussion has been also changed accordingly.**

3) I misss an analysis on the reliability of the proposed approach. Are LQ1-LQ6 the unique seismic signals with this characteristics? Or there area false positives or negatives? I think it would be nice to comment this in the work to understand how it works. Same happens with the location of the event. Is it the unique detected change? Or there are more? If there are more, how the authors atribute to the one selected? How many of the detected areas are landslides?

**This point has been raised also by reviewer 1. The idea of the paper is to show the feasibility of the combination of seismic data and remote sensing to have a better accuracy on the location and estimation of rock slope failure events. We have not performed a continuous processing of the seismic data to evaluate the proportion of false positive/false negatives, this beyond the scope and will be performed in further investigations. This is now specified in the text.**

Minor comments

- Line 147: "The outlier segment that identified covers ...... "  This sentence sounds strange to me.

**Thanks, corrected.**

Citation: https://doi.org/10.5194/nhess-2022-34-RC2